# Comorbidities of Rural Children and Adolescents with Migraine and without Migraine

**DOI:** 10.3390/children10071133

**Published:** 2023-06-29

**Authors:** Suzy Mascaro Walter, Zheng Dai, Kesheng Wang

**Affiliations:** 1Department of Family and Community Health, School of Nursing, West Virginia University, Morgantown, WV 26506, USA; kesheng.wang@hsc.wvu.edu; 2Health Affairs Institute, West Virginia University, Morgantown, WV 26506, USA; zheng.dai@hsc.wvu.edu

**Keywords:** rural, pediatric, adolescent, migraine, obstructive sleep apnea syndrome (OSAS), narcolepsy, restless leg syndrome (RLS), elevated blood pressure, depression, anxiety, Ehlers–Danlos Syndrome (EDS)

## Abstract

(1) Background: Migraine is associated with comorbidities that are common in the general rural pediatric population. The purpose of this study is to evaluate the differences in the occurrence of comorbidities between rural children and adolescents with and without migraine. (2) Methods: A cross-sectional, secondary data analysis using electronic medical records of 1296 patients (53.8% females, aged 12.4 ± 3.2) was completed. Mann–Whitney U test was used to detect the difference in the number of comorbidities between the two groups. Chi-square test was used to identify the differences in the number of comorbidities, which were classified as low (0–1 comorbidities), medium (2–3 comorbidities), and high (4 or plus comorbidities) degree of comorbidities. (3) Results: Significant differences were found between those children and adolescents with migraine vs. those without for depression (*p* < 0.0001), anxiety (*p* < 0.0001), and Ehlers–Danlos Syndrome (EDS; *p* = 0.0309). A marginally significant difference was found between those children and adolescents with migraine (47.2%; *n* = 306) vs. those without (42.1%; *n* = 273) for unhealthy weight (*p* = 0.0652). Approximately 40% of the migraineurs had 2–3 comorbidities, whereas 32% of the non-migraineurs had 2–3 comorbidities (*p* = 0.0003). (4) Conclusions: Findings demonstrate the importance of identifying comorbidities associated with rural pediatric migraine in order to develop effective treatment strategies that optimize patient outcomes.

## 1. Introduction

Migraine is a common chronic neurological disorder in rural children and adolescents and is associated with multiple comorbidities, including obesity, sleep disorders, hypertension, depression, anxiety, as well as musculoskeletal pain [1,2]. Rural children and adolescents have a higher prevalence of being overweight/obese, and comorbidities associated with childhood and adolescent obesity include hypertension, depression, anxiety, and sleep disorders [1,3,4]. Additionally, lower socioeconomic status is associated with higher pain prevalence, including headache, as well as musculoskeletal pain that can be attributed to joint hypermobility (i.e., Ehlers–Danlos Syndrome; EDS) [2,5]. Since the above diagnoses are comorbid with migraine and commonly occur in rural populations, it is important to monitor each of these comorbidities in rural pediatric patients who present with migraine.

The relationship between obesity and migraine is thought to be multifactorial, with the involvement of lifestyle behaviors as well as central and peripheral pathways that regulate feeding and overlap with pathways responsible for migraine [6,7,8]. For example, migraine triggers and prodromal symptoms (i.e., food cravings) have been linked to the role of the hypothalamus in pain modulation [9]. Migraine also shares a “complex yet poorly understood” relationship with sleep disorders, including obstructive sleep apnea syndrome (OSAS), narcolepsy, and restless leg syndrome (RLS) [10]. Migraine and sleep problems are thought to be associated with shared anatomical pathways. For example, the hypothalamus has a central role in the pathophysiology of migraine and as a regulator of sleep [10].

Despite the increasing prevalence of hypertension in children and adolescents, only 26% of children with documented blood pressures consistent with hypertension are diagnosed [11,12]. Symptoms that have been associated with hypertension in children and adolescents include fatigue, headache, and visual disturbances [13]. Headaches attributed to hypertension are characterized as bilateral and pulsating and are typically associated with an acute rise in systolic (to >180 mm Hg) and/or diastolic (to >120 mm Hg) blood pressure [14]. Symptoms associated with primary hypertension in children and adolescents include headaches and cognitive changes [15]. Headache remission occurs upon normalization of blood pressure [16]. Headaches associated with secondary hypertension may also be related to pheochromocytoma, hypertensive crisis without hypertensive encephalopathy, hypertensive encephalopathy, preeclampsia or eclampsia, or autonomic dysreflexia [14]. 

Migraine and comorbidity with depression and anxiety each involve complicated mechanisms [17,18]. Mechanisms thought to be involved in migraine and depression comorbidity include abnormal brain development, genetics, neurotransmitters, sex hormones, as well as shared environmental factors and stress [18]. For anxiety, neurobiological mechanisms underlying the association with migraine include serotonergic dysfunction, dysregulation of the hypothalamic-pituitary-adrenal axis, hormonal influences, and psychological factors (i.e., pain-related cognition, avoidance learning, and anticipatory anxiety) [17]. 

EDS is one of the most common symptomatic joint hypermobility conditions seen in clinical practice [19]. Migraine is more prevalent among those with EDS, and EDS is considered a risk factor for migraine [20]. Joint hypermobility syndrome is a connective tissue disease characterized by joint instability and chronic musculoskeletal pain. EDS shares many of the clinical features of joint hypermobility syndrome and is considered by some as one disease process [21]. Although the mechanism underlying the association between EDS and migraine is not clear, one reported pathway appears to be associated with chronic joint hypermobility, which stimulates cervical afferents resulting in the activation of the trigeminal cervical complex and subsequent development of head pain [20]. 

Given the above, headache treatment plans need to focus not only on migraine but also on the comorbidities contributing to rural pediatric migraine and overall health. The specific aims of this study are to:Determine if there are any significant differences in the occurrence of comorbidities (unhealthy weight, OSAS, narcolepsy, RLS, elevated blood pressure, depression, anxiety, and EDS) between rural children and adolescents with and without migraine.Examine if there is a significant difference in the mean number of comorbidities between rural children and adolescents with and without migraine.

## 2. Methods

### 2.1. Design

This is a cross-sectional study using secondary data obtained from an electronic medical record (EMR). 

### 2.2. Setting and Sample

Approval for this study was obtained from the Institutional Review Board (IRB) as exempt. Data were obtained from the Integrated Data Repository (IDR), a comprehensive patient database that collects patient information from the West Virginia University (WVU) Medicine Epic Clarity system, housed at the West Virginia Clinical and Translational Science Institute (WVCTSI). A data analyst from the WVCTSI retrieved EMR using Structured Query Language (SQL) from pediatric patients aged 7–17 who were evaluated in a rural pediatric outpatient clinic between 1 December 2009 to 31 December 2017. The clinic is part of an academic medical center in rural North Central West Virginia. All patients’ records were de-identified from the extracted data.

The overall sample included 1296 patients. The migraineurs were identified from those who were evaluated for headaches and had a primary diagnosis of migraine based on classification standards [16]. In contrast, the non-migraineurs were selected from those who were never diagnosed with any type of headache during the study period. Comorbidities included unhealthy weight, OSAS, narcolepsy, RLS, elevated blood pressure, depression, anxiety, and EDS, which were analyzed and compared between the two groups. 

### 2.3. Measures

#### Gender, Age, Race, and Ethnicity

Demographic factors included participants’ age (7–9 years, 10–12 years, and 13–17 years), sex (male and female), and race/ethnicity (White, Black, and Other).

### 2.4. Unhealthy Weight (Overweight and Obese)

Age, weight, height, sex, and body mass index (BMI) percentiles were obtained from the EMR. Overweight and obese were defined as a BMI of greater than or equal to the age- and sex-specific 85th percentile and 95th percentile, respectively, based upon the 2000 Centers for Disease Control and Prevention growth charts [22]. BMI was calculated using weight (lb)/[height (in)]^2^ × 703. BMI percentiles were computed and included the following categories: 1 = underweight (less than the 5th percentile), 2 = healthy weight (5th percentile to less than the 85th percentile), 3 = overweight (85th to less than the 95th percentile), and 4 = obese (equal to or greater than the 95th percentile) [22]. For the purposes of this study, unhealthy weight will refer to those children and adolescents who are overweight or obese. 

### 2.5. Other Comorbidities

The EMR charts were audited to identify the presence of the following diagnosis for both migraineurs and non-migraineurs: OSAS, narcolepsy, RLS, depression, anxiety, and EDS by ICD-10 codes or ICD-9 codes in the EMR. Elevated blood pressure in youth was determined by EMR-documented systolic and diastolic blood pressure, and categorization was based on a simplified blood pressure (BP) screening tool [23]. The term elevated blood pressure refers to isolated measurements and not to the diagnosis of hypertension.

### 2.6. Statistical Analysis

Descriptive analyses included frequency and percentages of demographics (gender, age, race, and ethnicity) and covariate comorbidities (elevated blood pressure, OSAS, narcolepsy, RLS, depression, anxiety, unhealthy weight, and EDS). The chi-square test was used to identify the differences in covariates between migraine and non-migraine groups. The number of comorbidities was considered a continuous variable and was presented in the form of mean ± standard deviation (SD). The Shapiro test was used to test normality on the number of comorbidities (*n* = 1296 which is smaller than 2000) and, since the p-value was significant (*p* < 0.0001), the null hypothesis of normal distribution was rejected. Then, a Mann–Whitney U test was performed to detect the difference in comorbidities between the two groups. Chi-square test was used to identify the differences in the number of comorbidities, which were classified as either one or multiple comorbidities, or low (0–1 comorbidities), medium (2–3 comorbidities), and high (4 or plus comorbidities) degree of comorbidities.

Secondary analyses compared the frequency and percentages of comorbidities among children and adolescents categorized as overweight/obese between migraine and non-migraine groups. The distribution of BMI and weight status between children and adolescents with and without migraine was also analyzed. Outcomes of interest include count and percent of each weight category (underweight, healthy weight, overweight, or obese) and associated mean, SD, and range of BMI in each weight category. 

Bivariate and multivariable logistic regression models were used to examine the risk factors associated with migraine among rural children and adolescents. The outcome was migraine, and the risk factors of interest included all covariates mentioned above, excluding those with a proportion lower than 1% of the overall population. Unadjusted odds ratios (uORs) and their 95% confidence intervals (CIs) were reported from bivariate logistic regression models, and adjusted odds ratios (aORs) and their 95% CIs were reported from the multivariable logistic regression model. Missing data were not included in the modeling. All statistical analysis was conducted using SAS (version 9.4; Cary, NC, USA). The significance level was set at 0.05, and all *p*-values were two-sided. To deal with the multiple-testing problem, the Bonferroni correction was used for statistical significance. Considering 11 independent variables, the Bonferroni corrected significant level will be a *p* value < 0.05/11 = 0.0045.

## 3. Results

Demographics and comorbidities of children and adolescents with and without migraine are presented in Table 1. Ethnicity comprised primarily White (95.8%), Black (1.7%), followed by other (2.5%) children and adolescents. Participants ranged in age from 7 to 17 years old, with the mean age being 12.4 (SD = 3.2; not shown in the table) years. The sample was 53.8% female. Age was categorized by 7–9 years old (23.5%), 10–12 years old (24.7%), and 13–17 years old (51.9%). There was a significant difference between the children and adolescents with migraine vs. those without migraine in terms of gender (*p* < 0.0001) and race (*p* = 0.0403). A greater number of females (61.4% vs. 38.6%) and a relatively smaller number of white children and adolescents (94.9% vs. 96.8%) were diagnosed with migraine. However, in terms of age, there was no significant difference between children and adolescents with migraine vs. those without migraine (*p* = 0.3329). Overall, comorbidities reported in this population (*n* = 1296) included depression (14.4%), anxiety (19.4%), unhealthy weight (44.7%), EDS (0.6%), elevated blood pressure (44.0%), OSAS (11.3%), narcolepsy (0.7%), and RLS(0.5%).

In those children and adolescents diagnosed with migraine, 20.5% (*n* = 133) and 28.2% (*n* = 183) were diagnosed with depression and anxiety, respectively (Table 1). Significant differences were found between those children and adolescents with migraine vs. those without for both depression (*p* < 0.0001) and anxiety (*p* < 0.0001). A marginally significant difference was found between those children and adolescents with migraine (47.2%; *n* = 306) vs. those without (42.1%; *n* = 273) for unhealthy weight (*p* = 0.0652). A significant difference was found between those children and adolescents with migraine vs. those without for EDS (*p* = 0.0309). Although the sample was small, approximately 1.1% (*n* = 7) of those with migraine were diagnosed with EDS vs. those without migraine (0.2%; *n* = 1). No significant difference in elevated blood pressure, OSAS, narcolepsy, or RLS between those with migraine vs. those without migraine was found. 

Given that the comorbidities described in this study co-occur in the overweight/obese child and adolescent population as well as the migraine population [1], Table 2 describes the prevalence of comorbidities in unhealthy-weight children and adolescents with and without migraine. There was no significant difference in elevated blood pressure, OSAS, narcolepsy, or RLS in overweight/obese children and adolescents with migraine vs. overweight/obese children and adolescents without migraine. However, a statistically significant difference was found in overweight/obese children and adolescents with migraine vs. without migraine for both depression (22.9% vs. 11.4%; *p* = 0.0003) and anxiety (29.1% vs. 12.1%; *p* < 0.0001).

Descriptive statistics of BMI and weight status between children and adolescents with or without migraine are presented in Table 3. The prevalence of obesity and overweight among children and adolescents with migraine was 26.4% and 20.8%, respectively, as compared to 26.2% and 15.9% in those without migraine. The distribution of weight categories between male and female were not statistically significantly different (*p* = 0.1980; not shown in the table). 

When looking at the average number of comorbidities, as shown in Table 4, there was a significantly greater number of comorbidities in those children and adolescents diagnosed with migraine 1.5 (SD = 1.2) compared with those without migraine 1.2 (SD = 1.0) (*p* < 0.0001 based on the Mann–Whitney U test). The analyses of the number of comorbidities suggest the proportion of cases with at least one comorbidity is significantly higher among children and adolescents with migraine vs. those without any comorbidities (*p* = 0.0002). Those with migraine have a statistically significantly higher percent of multiple comorbidities (compared with those with one comorbidity) and medium comorbidities (compared with those with low comorbidities) than non-migraine children and adolescents (*p* = 0.0048 and 0.0003, respectively). Additionally, those with migraine have a marginally statistically significantly higher percent of high comorbidities (compared with those with medium comorbidities) than non-migraine children and adolescents (*p* = 0.0728). 

Table 5 summarizes the estimated model effects (uOR and aOR with 95% CIs) of factors associated with the outcome variable. In bivariate models, being male had a protective factor associated with a lower rate of migraine (vs. female, uOR = 0.54; 96% CI = 0.43–0.67). Risk factors included race being “other” than white or black (vs. white, 2.60; 1.20–5.67), depression (vs. no depression, 2.84; 2.03–3.98), and anxiety (vs. no anxiety, 3.36; 2.45–4.55). All factors were significantly associated with migraine. After adjusting for all demographics and comorbidities in the multivariable model, the significant associations held up with slightly different effects size, male (0.60; 0.48–0.76), race being “other” than white or black (2.70; 1.21–6.00), depression (1.68; 1.15–2.47), and anxiety (2.66; 1.90–3.71). Specifically, compared to a healthy weight, unhealthy weight was marginally statistically associated with migraine, with uOR of 1.23 (0.99, 1.53) and aOR of 1.15 (0.91, 1.47).

## 4. Discussion

This is the first study to evaluate comorbidities in children and adolescents with and without migraine in a rural pediatric population. Approximately 61% of female children and adolescents were diagnosed with migraine as opposed to 38.6% of males (*p* < 0.0001). In bivariate logistic models, the results of this study demonstrated being male is a protective factor. These findings are similar to other studies that report a higher prevalence of headaches in female (26%) vs. male (19%) adolescents [4]. Factors including premenstrual syndrome, menses, use of birth control, and hormones (e.g., prolactin) may contribute to the female predominance of headaches [24,25]. 

In bivariate logistic models, this study also found that race, depression, and anxiety were considered risk factors. Race in this study, being “other” than black or white, was considered a risk factor. Given the nature of secondary data analysis, “other” was not clearly defined in terms of ethnicity and/or race. In other studies that evaluated migraine, the prevalence was higher in White than in Hispanic, African American, and Asian adolescents and adults [4,26]. The lack of reported findings in varied ethnic groups in this study is due to a lack of diversity in rural WV, as evidenced by a racial/ethnic mix of 92% White, 4% Black, 0% Native American, 1% Asian, 0% other, and 1% Hispanic [27]. Further research should investigate urban populations in order to make comparisons for migraine comorbidities between rural and urban pediatric groups.

EDS is associated with chronic pain (musculoskeletal and non-musculoskeletal), sleep disorders, mood and anxiety disorders, alongside fatigue, orthostatic intolerance, and functional gastrointestinal disorders [19,28,29,30]. Specifically, in terms of non-musculoskeletal pain, headache is frequently reported in those with EDS [5,28,30]. Additionally, comorbidities in EDS overlap with comorbidities in adolescent headaches (i.e., depression, anxiety, and sleep disorders). In this study, a significant difference was found between those children and adolescents with migraine vs. those without for EDS. However, the sample was small; approximately 1.1% (*n* = 7) of those with migraine were diagnosed with EDS. Diagnosis of EDS and comorbidities are often not recognized by health care professionals, leading to significant delays in diagnosis [31]. Thus, the representation of EDS in this study may be due to missed diagnoses in this rural migraine population. 

Depression and anxiety are well-known comorbidities with childhood and adolescent headaches [32,33]. In a nationally representative sample of US youth, adolescent headaches and migraine were significantly associated with anxiety, mood, and behavior disorders [34]. Additionally, youth with headaches were found to be twice as likely to meet the criteria for a DSM-IV disorder, including depression and anxiety [34]. In this study, children and adolescents with migraine had over a 2.5 times higher proportion of depression and anxiety compared to those without migraine. Significant differences were found between those children and adolescents with migraine vs. those without for both depression (*p* < 0.0001) and anxiety (*p* < 0.0001). 

Approximately 47% of children and adolescents in rural West Virginia are either overweight or obese [35]. In this study, approximately 44.7% (*n* = 1296) of rural children and adolescents were overweight or obese. A marginally significant difference was found between those children and adolescents with migraine vs. those without for those in the unhealthy weight group (*p* = 0.0652). In the bivariate logistic model, unhealthy weight was marginally statistically associated with migraine, with uOR of 1.23 (0.99, 1.53) and aOR of 1.15 (0.91, 1.47) compared to healthy weight. Pediatric studies have reported significant associations between headaches and obesity [36,37]. The relationship is likely multifactorial and influenced by environmental risk factors (i.e., poor physical activity), genetic factors, biochemical markers (orexins, adipose tissue function, and estrogen levels), as well as psychologic factors (depression and anxiety) [7]. Obesity is considered a modifiable risk factor for migraine in the pediatric population [38]. Further research should evaluate weight management strategy interventions that contribute to the improvement of both migraine and BMI outcomes for overweight/obese rural adolescents [1]. Further, this study found a statistically significant difference in overweight/obese children and adolescents with vs. without migraine for both depression (*p* = 0.0003) and anxiety (*p* < 0.0001). These findings provide evidence of the importance of integrating mental health in pediatric and adolescent headache weight management strategy interventions. 

Reports have shown that nearly a quarter (23.2%) of rural WV children meet the criteria for hypertension [35]. In this study, approximately 44% (*n* = 560) met the criteria for elevated blood pressure [23]. Of those, 47.5% were diagnosed with migraine. However, there was no significant difference between those children and adolescents with migraine vs. those without for elevated blood pressure (*p* = 0.3247). In another study, findings also demonstrated no significant association between elevated blood pressure and any headache disorder (migraine, tension-type headaches, chronic daily headaches) [39]. In this study, only elevated blood pressure was reported, no distinction was made for the stages of hypertension, nor could “white coat hypertension” be ruled out. Ambulatory blood pressure monitoring (ABPM) has been used to better assess for white coat hypertension or masked hypertension in children with elevated blood pressure [13]. Future studies using ABPM and current diagnostic criteria can also be used to examine the association between blood pressure and headaches in children and adolescents [40]. 

Sleep related comorbidities, associated with migraine, that are reported in this study population included OSAS (11.3%), narcolepsy (0.7%), and RLS (0.5%). The prevalence of OSAS has been reported at 40–56.5% in children and adolescents with migraine [41,42]. In this study, only 12.0% (*n* = 78) of the children and adolescents with migraine had a diagnosis of OSAS as compared with 10.5% (*n* = 68) of those without migraine. These findings may be due to missed diagnoses of sleep disorders in the migraine population, as there is a general consensus that the prevalence of sleep-disordered breathing is underestimated in the childhood and adolescent population [43]. 

Migraine has been reported as an independent risk factor for narcolepsy development in children [44]. In this study, no significant differences were found between those children and adolescents with migraine vs. those without for narcolepsy (*p* = 0.7380). However, only five children with migraine (0.8%) and four children without migraine (0.6%) were diagnosed with narcolepsy. Additionally, no significant differences were found between those children and adolescents with migraine vs. those without for RLS (*p* = 1.00). This is in contrast to previous findings that have demonstrated a significantly higher frequency of RLS in pediatric migraine patients compared to controls [45]. Again, these findings may be due to missed diagnoses of both narcolepsy and RLS in children and adolescents. For example, more than 50% of adults report symptoms of narcolepsy during adolescence; however, the delay between symptom onset and diagnosis of narcolepsy can range from 10–15 years [46]. 

There was a significant difference in the mean number of comorbidities between those children and adolescents diagnosed with migraine and those without migraine (*p* < 0.001). The mean number of comorbidities was higher in children and adolescents with migraine versus vs. those without. Approximately 47% of the migraineurs had multiple comorbidities, whereas 35% of the non-migraineurs had multiple comorbidities (*p* = 0.0048). Approximately 40% of the migraineurs had two to three (medium) comorbidities, whereas 32% of the non-migraineurs had two to three comorbidities (*p* = 0.0003). 

These findings demonstrate the importance of understanding the relationship between migraine and associated comorbidities in a rural pediatric population. Once the diagnosis of migraine is determined, a thorough assessment is necessary in order to address the presence of multiple comorbidities that may be contributing to recurrent or chronic headaches.

### 4.1. Strengths and Limitations

Strengths of this study include a large sample size representative of a geographically distinct rural clinic. Migraine diagnosis was determined by specialists in pediatric and adolescent headache using classification standards [16]. Limitations include reliance on diagnosis codes for comorbidities; thus, the risk of over- or under-reporting may occur. For example, it is unknown if valid and reliable instruments were consistently used to diagnose depression and anxiety. Additionally, it is unknown if these children and adolescents were appropriately evaluated for the comorbidities reported to be associated with primary headaches (i.e., sleep disorders, EDS). Medication interventions have the potential to impact headaches, weight, mood, and sleep; however, this review did not include medication in the data collection. Diagnosis of hypertensive blood pressure could not be made due to the isolated measurements of this cross-sectional study. Since this was a cross-sectional study design, causality could not be addressed. Finally, the use of a rural clinic-based sample limits the generalizability of findings.

### 4.2. Implications

This study found that the mean number of comorbidities was higher in the children and adolescents with migraine vs. those without. Additionally, significant differences were found between those children and adolescents with migraine vs. those without for depression, anxiety, and EDS, as well as approaching significance for unhealthy weight. Further, this study found a statistically significant difference in overweight/obese children and adolescents with vs. without migraine for both depression and anxiety.

The above findings underscore the importance of monitoring comorbidities in rural children and adolescents with migraine. When evaluating rural children and adolescents for headaches, evaluation and treatment of migraine must involve an approach that includes assessment of comorbidities. Failure to recognize and treat comorbidities may result in poorer headache prognosis. For example, undiagnosed or untreated sleep disorders place the adolescent at risk for increased frequency and severity of migraine and possible headache chronification [47]. The burden of delayed diagnosis in EDS may not only contribute to poorer headache outcomes but also morbidity and disability due to poorly treated joint hypermobility [31]. Healthcare providers should assess and record adolescent blood pressure and monitor elevated blood pressure for possible hypertension. The association of hypertension in pediatric and adolescent primary headaches is important to evaluate; however, hypertension may also serve as an indicator for secondary causes of headaches (i.e., hypothyroidism, hyperparathyroidism) [48,49]. Finally, undiagnosed mental health disorders increase the chronic nature of headaches, make headaches less responsive to treatment and contribute to the persistence of headache-related disability [1].

## 5. Conclusions

The prevalence of primary headaches in children and adolescents has been reported at 62%; however, pediatric headaches are considered to be an underdiagnosed and undertreated form of pain in this population [50]. Poorly treated headaches in children and adolescents negatively affect health-related quality of life (HrQoL), school attendance and performance, as well as social functioning [51,52]. Poor headache management in children and adolescents leads to persistent headaches into adulthood; thus, providers should be educated on proper headache management in the pediatric population [53]. 

Headaches in rural children and adolescents are associated with multiple comorbidities that are common to the rural pediatric population. When undiagnosed or untreated, these comorbidities may contribute to both the onset and chronification of headache disorders. The above findings demonstrate the importance of monitoring these comorbidities in order to develop the most effective treatment strategies to optimize patient outcomes. Healthcare providers need to be aware of and appropriately assess comorbidities associated with pediatric and adolescent migraine to optimize headache and overall health outcomes. 

## Figures and Tables

**Table 1 children-10-01133-t001:** Demographics and co-morbidities of rural children and adolescents with migraine and without migraine.

Characteristics	All, N (%)	Non-Migraineurs, N (%)	Migraineurs, N (%)	*p*-Value
	(N = 1296)	(N = 648)	(N = 648)	
Age				0.3329
7–9 years	304 (23.5)	163 (25.2)	141 (21.8)	
10–12 years	320 (24.7)	159 (24.5)	161 (24.9)	
13–17 years	672 (51.9)	326 (50.3)	346 (53.4)	
Gender				<0.0001 *
Male	599 (46.2)	349 (53.9)	250 (38.6)	
Female	697 (53.8)	299 (46.1)	398 (61.4)	
Race				0.0403
White	1242 (95.8)	627 (96.8)	615 (94.9)	
Black	22 (1.7)	12 (1.9)	10 (1.5)	
Other	32 (2.5)	9 (1.4)	23 (3.5)	
Comorbidities				
Elevated BP	560 (44.0)	294 (45.4)	266 (42.6)	0.3247
OSAS	146 (11.3)	68 (10.5)	78 (12.0)	0.3796
Narcolepsy	9 (0.7)	4 (0.6)	5 (0.8)	0.7380
Restless leg syndrome	6 (0.5)	3 (0.5)	3 (0.5)	1.0000
Depression	187 (14.4)	54 (8.3)	133 (20.5)	<0.0001 *
Anxiety	251 (19.4)	68 (10.5)	183 (28.2)	<0.0001 *
Unhealthy weight	579 (44.7)	273 (42.1)	306 (47.2)	0.0652
EDS	8 (0.6)	1 (0.2)	7 (1.1)	0.0309

Note: BP (blood pressure); OSAS (obstructive sleep apnea syndrome); EDS (Ehlers Danlos Syndrome); all percentages are calculated by columns. * *p*-values are statistically significant after Bonferroni correction.

**Table 2 children-10-01133-t002:** Comparison of comorbidities of rural children and adolescents with unhealthy weight between migraineurs and non-migraineurs.

Characteristics	All, N (%)	Unhealthy Weight Non-Migraineurs, N (%)	Unhealthy Weight Migraineurs, N (%)	*p*-Value
	(N = 579)	(N = 273)	(N = 306)	
Comorbidities				
Elevated BP	328 (57.6)	167 (38.8)	161 (54.4)	0.1020
OSAS	90 (15.5)	42 (15.4)	48 (15.7)	0.9203
Narcolepsy	5 (0.9)	2 (0.7)	3 (1.0)	0.3294
Restless leg syndrome	6 (1.0)	3 (1.1)	3 (1.0)	0.3111
Depression	101 (17.4)	31 (11.4)	70 (22.9)	0.0003 *
Anxiety	122 (21.1)	33 (12.1)	89 (29.1)	<0.0001 *
EDS	0 (0.0)	0 (0.0)	0 (0.0)	NA

Note: unhealthy weight includes overweight and obesity; BP (blood pressure); OSAS (obstructive sleep apnea syndrome); EDS (Ehlers Danlos Syndrome); all percentages are calculated by columns. * *p*-values are statistically significant after Bonferroni correction.

**Table 3 children-10-01133-t003:** Descriptive statistics of BMI and weight status between children and adolescents with migraine and without migraine.

BMI	All		Non-Migraineurs		Migraineurs	
	N, (%)	Mean (SD) [Range]	N, (%)	Mean (SD) [Range]	N, (%)	Mean (SD) [Range]
Underweight	40 (3.1)	14.7 (1.5) [12.1, 17.5]	28 (4.3)	14.6 (1.5) [12.1, 17.5]	12 (1.9)	15.0 (1.4) [13.2, 17.4]
Healthy Weight	677 (52.2)	18.5 (2.6) [13.7, 25.2]	347 (53.6)	18.2 (2.5) [13.8, 25.2]	330 (50.9)	18.8 (2.7) [13.7, 25.1]
Overweight	238 (18.4)	23.3 (2.9) [17.4, 29.4]	103 (15.9)	23.5 (2.6) [17.8, 28.7]	135 (20.8)	23.1 (3.1) [17.4, 29.4]
Obese	341 (26.3)	30.2 (6.2) [19.4, 58.5]	170 (26.2)	29.8 (6.0) [19.4, 51.7]	171 (26.4)	30.6 (6.5) [20.0, 58.5]

**Table 4 children-10-01133-t004:** Comparison in number of comorbidities between children and adolescent with migraine and without migraine.

Number of Comorbidities ^^^	All, N (%)	Non-Migraineurs, N (%)	Migraineurs, N (%)	*p*-Value
Mean (SD)	1.3 (1.1)	1.2 (1.0)	1.5 (1.2)	<0.0001 *
0	331 (25.5)	195 (30.1)	136 (21.0)	
1	433 (33.4)	225 (34.7)	208 (32.1)	
2	355 (27.4)	165 (25.5)	190 (29.3)	
3	115 (8.9)	43 (6.6)	72 (11.1)	
4	53 (4.1)	19 (2.9)	34 (5.3)	
5	8 (0.6)	1 (0.2)	7 (1.1)	
6	1 (0.1)	0 (0.0)	1 (0.2)	
At least one comorbidity	965 (74.5)	453 (69.9)	512 (79.0)	0.0002 ^a^*
Multiple comorbidities	532 (41.0)	228 (35.2)	304 (46.9)	0.0048 ^b^
Low comorbidities	764 (59.0)	420 (64.8)	344 (53.1)	NA
Medium comorbidities	470 (36.3)	208 (32.1)	262 (40.4)	0.0003 ^c^*
High comorbidities	62 (4.8)	20 (3.1)	42 (6.5)	0.0728 ^d^

^a^ compares the number of cases with no comorbidity vs. at least one comorbidity between migraineurs and non-migraineurs. ^b^ compares the number of cases with only one comorbidity vs. multiple comorbidities between migraineurs and non-migraineurs. ^c^ compares the number of cases with zero or one comorbidity vs. two or three comorbidities between migraineurs and non-migraineurs. ^d^ compares the number of cases with two or three comorbidities vs. more than three comorbidities between migraineurs and non-migraineurs. ^^^ Comorbidities include hypertension, OSAS (obstructive sleep apnea syndrome), narcolepsy, RLS (restless leg syndrome), depression, anxiety, unhealthy weight, and EDS (Ehlers Danlos Syndrome), SD (standard deviation). * *p*-values are statistically significant after Bonferroni correction.

**Table 5 children-10-01133-t005:** Unadjusted and adjusted odds ratios for associations between factors and migraine among children and adolescents.

Variable	uOR [95% CI]	aOR [95% CI]
Age		
7–9 years	1	1
10–12 years	1.17 [0.86, 1.60]	1.11 [0.80, 1.55]
13–17 years	1.23 [0.94, 1.61]	1.02 [0.76, 1.36]
Gender		
Female	1	1
Male	0.54 [0.43, 0.67]	0.60 [0.48, 0.76]
Race		
White	1	1
Black	0.85 [0.36, 1.98]	0.75 [0.29, 1.92]
Other	2.60 [1.20, 5.67]	2.70 [1.21, 6.00]
Comorbidities		
Elevated BP	0.90 [0.72, 1.12]	0.92 [0.72, 1.17]
OSAS	0.86 [0.61, 1.21]	1.06 [0.73, 1.53]
Depression	2.84 [2.03, 3.98]	1.68 [1.15, 2.47]
Anxiety	3.36 [2.45, 4.55]	2.66 [1.90, 3.71]
Unhealthy weight	1.23 [0.99, 1.53]	1.15 [0.91, 1.47]

Note: uOR unadjusted odds ratio; aOR adjusted odds ratio; BP (blood pressure); OSAS (obstructive sleep apnea syndrome).

## Data Availability

The data are not publicly available due to due to privacy and ethical restrictions. Data were collected from a patient database that is not publicly available.

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
