# Peer review of "Comorbidities of Rural Children and Adolescents with Migraine and without Migraine"

_children, 2023, doi:10.3390/children10071133_

Round 1

Reviewer 1 Report

The paper presented to me for review deals with the very important issue of comorbidities of rural children and adolescents with migraine and without migraine. This is an research paper that addresses a very important practical issue that is rarely analyzed in the child population.

The results suggest that children with migraine have a higher health burden of psychiatric illnesses in particular. The paper is written in good and clear language. The results are presented clearly. However, before accepting the paper for publication, I would suggest supplementing it with a few points:

1. it would be useful to refer to a recent meta-analysis of the prevalence of headaches in children, which suggests that it is a far greater problem than is commonly believed: PMCID: PMC9926688

2. in the introduction, the sequence should be corrected: "Migraine is a common chronic pain syndrome" because migraine is not a symptom but a neurological disease. 

3. in the discussion, reference should be made to a paper showing another important aspect of the development of the disease: it has been shown in the adult population that the onset of migraine is most common in childhood and not, as previously thought, in young adults: PMID: 37041492. moreover, this work shows similarities in comordibities in adult and pediatric populations 

Reviewer 2 Report

MS Title: Co-morbidities of rural children….

MS No: children-2443668

MS Authors: Walter et al.

Date of review: 2023-06-08

In this retrospective cross-sectional secondary data analysis study, the authors studied the association of migraine with some other comorbidities in the general rural pediatric population. Such association in the pediatric population has also been studied before, but not many data were obtained from a particular socioeconomic regions (e.g., rural area in certain states in USA). Therefore, the purpose (and also the limitation) of this study is to evaluate the differences in the occurrence of comorbidities between rural children and adolescents with and without migraine. The study is well designed and conducted. The results are well analyzed and presented. In conclusion, the authors identified a number of comorbidities  associated with rural pediatric migraine cases and expect to provide a way to develop effective treatment strategies that optimize patient outcomes in the future.

Minor comments:

1.      Page 8, line 252: It would be better here to cite the number of prevalence rate in the reference-4, namely “26%”.

2.      Page 8, line 263: If the racial/ethnic mix ratio indicating 92% of the study subjects are White, would it be better to indicate such ethnic (or racial) predominance in the title of this article?

Author Response

Thank you for your thoughtful and thorough review of our manuscript, please see responses below to your comments as well as highlights in the manuscript:

Reviewer 2 Comments

Author’s Response

1.   Page 8, line 252: It would be better here to cite the number of prevalence rate in the reference-4, namely “26%”.

Prevalence rate was added to the following on page 8 LINE 261: These findings are similar to other studies that report a higher prevalence of headache in female (26%) vs. male (19%) adolescents [4].

2.      Page 8, line 263: If the racial/ethnic mix ratio indicating 92% of the study subjects are White, would it be better to indicate such ethnic (or racial) predominance in the title of this article?

Since our study did not limit our analysis to White rural children and adolescents, we prefer to not use White in the title of this article. Especially since  Race, being “other” than black or white, in this study, was considered a risk factor. Also, per the NIH, to account for the diverse lived experiences and exposures of various populations, clinical research should be appropriately inclusive of racial and ethnic minority groups. Placing “white” in the title would infer that we did not attempt to account for other racial/ethnic groups.

Reviewer 3 Report

Thank you for your good report on rural children with/without migraine and comorbiditites.

Statistical problems have arisen.

Please check if the distribution is normal. If not, use MWUtest instead of t-test.

p<0.05 is not good. Please give a reason not to do the Bonferroni correction or do the Bonferroni correction.PMID: 2081237

Please think of a way to clearly differentiate between rural and urban areas. In particular, please consider with regard to the regional deprivation index.PMID: 25757802

  • PMID: 36724587 Also mention the significant disruption to life caused by migraine headaches in children. Also mention the need for further study of triggers.

Were there any missing values? Or did you have missing values and did you do multiple assignments, etc.? Were there really no missing values with all this data?

Discussion is very well put. I would suggest that you also mention the possibility of collecting urban data and making comparisons with urban areas in the future.

Author Response

Thank you for your thoughtful and thorough review of our manuscript, please see responses below to your comments as well as highlights in the manuscript:

Reviewer 3 Comments

Author’s Response

Please check if the distribution is normal. If not, use MWUtest instead of t-test.

In method page 3 LINES 154-159:

The number of comorbidities was considered as a continuous variable and was presented in the form of mean ± standard deviation (SD). After performing a normality test on the number of comorbidities using Shapiro test (n = 1296 which is smaller than 2000), the p-value < 0.0001, which rejects null hypothesis of normal distribution. Then a Mann-Whitney U test was performed to detect the difference of comorbidities between the two groups.

In results page 6 LINE 232:

(p<0.0001 based on the Mann-Whitney U test)

In the abstract page 1 LINE 16:

Mann-Whitney U test was used to detect the difference of number of comorbidities be-tween the two groups.

p<0.05 is not good. Please give a reason not to do the Bonferroni correction or do the Bonferroni correction.PMID: 2081237

In method page 4 LINES 178-181:

To deal with the multiple testing problem, Bonferroni correction was used for statistical significance. Considering 11 independent variables, the Bonferroni corrected significant level will be a p value < 0.05/11=0.0045.

Please think of a way to clearly differentiate between rural and urban areas. In particular, please consider with regard to the regional deprivation index.PMID: 25757802

As described in the Setting and Sample, a data analyst retrieved data from a rural pediatric outpatient clinic in North Central West Virginia.

West Virginia is considered a part of Appalachia and the entire state is rural so there is no need to differentiate any part of WV between rural vs. urban- there are no urban areas in West Virginia. We appreciate the importance of the neighborhood deprivation, and would consider this a separate study for future work.

PMID: 36724587 Also mention the significant disruption to life caused by migraine headaches in children.

Also mention the need for further study of triggers.

The following was added on page 11 LINE 399: Poorly treated headache in children and adolescents negatively effects health-related quality of life (HrQoL), school attendance and performance, as well as social functioning [52,53].

The authors would like to defer mention of triggers. Although headache triggers are a significant part of evaluation for migraine, our article is focused on co-morbidities and not specific characteristics of migraine workup (ie. triggers).

Were there any missing values? Or did you have missing values and did you do multiple assignments, etc.? Were there really no missing values with all this data?

Data was drawn from electronic medical records from a clinical setting. All data used in this study are routinely documented as this information is required for the provider to  assess the patient. Electronic documentation that is a requirement for each visit includes, for example, blood pressure, weight and height, each of which is assessed and documented at each clinic appointment. Electronic documentation always has available the age and sex of the patient as well as the race/ethnicity. Thus, it is not expected to have missing values for gender, age, race/ethnicity. Nor will there be missing data for weight or blood pressure as these are a requirement for each patient visit and are documented in the electronic medical record. Thus, there were no missing values.

Discussion is very well put. I would suggest that you also mention the possibility of collecting urban data and making comparisons with urban areas in the future.

In the discussion, page 8, LINE 273, the following was added: Further research should investigate urban populations in order to make comparisons for migraine comorbidities between rural and urban pediatric groups.